# Farnesol Boosts the Antifungal Effect of Fluconazole and Modulates Resistance in *Candida auris* through Regulation of the *CDR1* and *ERG11* Genes

**DOI:** 10.3390/jof8080783

**Published:** 2022-07-27

**Authors:** Jaroslava Dekkerová, Lucia Černáková, Samuel Kendra, Elisa Borghi, Emerenziana Ottaviano, Birgit Willinger, Helena Bujdáková

**Affiliations:** 1Department of Microbiology and Virology, Faculty of Natural Sciences, Comenius University in Bratislava, Ilkovicova 6, 84215 Bratislava, Slovakia; jaroslava.dekkerova@uniba.sk (J.D.); lucia.cernakova@uniba.sk (L.Č.); kendra4@uniba.sk (S.K.); 2Department of Health Sciences, San Paolo Medical School, Università Degli Studi di Milano, Via A. di Rudini 8, 20142 Milan, Italy; elisa.borghi@unimi.it (E.B.); emerenziana.ottaviano@unimi.it (E.O.); 3Division of Clinical Microbiology, Department of Laboratory Medicine, Medical University of Vienna, 1090 Vienna, Austria; birgit.willinger@meduniwien.ac.at

**Keywords:** *Candida auris*, antifungal resistance, farnesol, efflux pumps

## Abstract

*Candida auris* is considered a serious fungal pathogen frequently exhibiting a high resistance to a wide range of antifungals. In this study, a combination of the quorum-sensing molecule farnesol (FAR) and fluconazole (FLU) was tested on FLU-resistant *C. auris* isolates (*C. auris* S and *C*. *auris* R) compared to the susceptible *C. auris* H261. The aim was to assess the possible synergy between FAR and FLU, by reducing the FLU minimal inhibitory concentration, and to determine the mechanism underlying the conjunct effect. The results confirmed a synergic effect between FAR and FLU with a calculated FIC index of 0.75 and 0.4 for *C. auris* S and *C. auris* R, respectively. FAR modulates genes involved in azole resistance. When FAR was added to the cells in combination with FLU, a significant decrease in the expression of the *CDR1* gene was observed in the resistant *C. auris* isolates. FAR seems to block the Cdr1 efflux pump triggering a restoration of the intracellular content of FLU. These results were supported by observed increasing accumulation of rhodamine 6G by *C. auris* cells. Moreover, *C. auris* treated with FAR showed an *ERG11* gene down-regulation. Overall, these results suggest that FAR is an effective modulator of the Cdr1 efflux pump in *C. auris* and, in combination with FLU, enhances the activity of this azole, which might be a promising strategy to control infections caused by azole-resistant *C. auris.*

## 1. Introduction

Finding an effective tool to combat the multidrug-resistant pathogen *Candida auris* is a challenge indeed. Isolated for the first time in 2009 in Japan, this fungus is currently widespread worldwide [1,2,3]. *C. auris* colonizes the skin and persists in the healthcare environment, frequently resulting in invasive outbreaks due to its capability to form adherent biofilms on clinically relevant surfaces [3,4,5]. Despite the increased incidence of *C. auris* infections, diagnostic tools remain difficult because of the lack of available conventional approaches. Additionally, therapeutic options are still limited [2,6]. *C. auris* isolates manifest resistance to common antifungals such as fluconazole (FLU) and amphotericin B (AMB) [7,8]. Moreover, resistance to echinocandins is emerging in some countries and has been shown to be acquired rather than intrinsic [4]. On the other hand, concomitant resistance to all three classes of antifungals (azoles, AMB, and echinocandins) is rare [3].

Overregulation of efflux pumps such as ATP binding cassette (ABC) and major facilitator superfamily (MFS) are known to cause multidrug resistance in *Candida* species [9]. Efflux together with mutations in the azole target gene *ERG11* have been identified as the main mechanisms of resistance in *C. auris* [10,11]. Using a combination of antifungal drugs is a well-established approach for the treatment of several fungal infections, but to date, there is little evidence to support combined therapy in *C. auris* [12]. Searching for molecules able to modulate the efflux pumps synergizing with azoles against *C. auris* is a highly topical issue [13].

Farnesol (FAR) is a fungal quorum-sensing molecule that has been demonstrated in *C. albicans* to enhance the activity of antifungals and to affect the ABC efflux transporters resulting in decrease in the azole resistance [10,14]. The other favorable effect of FAR in combination with echinocandins (caspofungin and micafungin) was determined against *Candida parapsilosis* biofilms [15]. Another study showed the prominent effect of FAR with echinocandins against *C. auris* biofilms as well [16]. Furthermore, FAR blocked efflux pumps and down-regulated genes involved in biofilm development and resistance [5]. Modulation of *C. auris* efflux pump activity by FAR could represent a promising approach for controlling life-threatening infections caused by this pathogen [5,10]. Concerning obtained information, FAR should have a potential adjuvant effect in the eradication of resistant *C. auris.*

The aim of the presented study was to investigate the possible effect of the combination of FAR and FLU on the expression of the genes relevant to FLU resistance in *C. auris* strains manifesting different FLU susceptibility profiles. 

## 2. Materials and Methods

### 2.1. Characterization of the C. auris Strains

Three isolates of *C. auris* were used in this study: *C. auris* H261 was kindly provided by Prof. Birgit Willinger (Medical University, Vienna, Austria). The strain was isolated in January 2018 from a healthy 22-year-old man with therapy-refractory external otitis. The detailed characterization, including the susceptibility profile, was published by Pekard-Amenitsch et al. (2018) [17]. The strain was deposited in the CBS-KNAW yeast collection of the Westerdijk Fungal Biodiversity Institute in Utrecht, the Netherlands, and assigned as CBS 15366. The isolates *C. auris* S and *C. auris* R were kindly provided by Prof. Maurizio Sanguinetti (Università Cattolica del Sacro Cuore, Rome, Italy). The clinical isolates were isolated from patients with positive blood cultures in hospitals in India. Strains were originally marked as *C. auris* 1a and *C. auris* 45 [18]. In the presented paper, the names have been changed due to different susceptibility profiles to FLU. The FLU-susceptible strain *C. auris* 1a was renamed as *C. auris* S and FLU-resistant strain *C. auris* 45 was renamed as *C. auris* R.

All strains were preserved at −80 °C in 1 mL of yeast extract peptone dextrose broth (YPD broth; 1% yeast extract, Biolife, Italy; 2% mycological peptone, Lab M Limited, Buri, UK; 2% D-glucose, Centralchem, Bratislava, Slovakia) supplemented with 60% glycerol (Centralchem, Bratislava, Slovakia). Prior to experiments, yeast cells from stocks stored at −80 °C were streaked onto a yeast extract–peptone–dextrose plate (YPD) supplemented with 2% agar (Biolife, Milan, Italy) and incubated overnight at 30 °C. One loopful of cells from YPD agar plates was inoculated into flasks containing 20 mL of YPD broth and grown in an orbital shaker (Multitrone Standard, Infors HT, Bottmingen-Basel, Switzerland), at 180 rpm for up to 16 h at 30 °C. The cells were then washed twice with phosphate-buffered saline (PBS) buffer (Sigma, St. Louis, MO, USA) and adjusted to the appropriate density for each experiment.

The susceptibility profile of *C. auris* isolates was determined by the broth microdilution method according to the European Committee on Antimicrobial Susceptibility Testing (EUCAST) protocol [19] in Roswell Park Memorial Institute medium (RPMI 1640 medium, Biowest, Nuaillé, France) supplemented with 2% D-glucose (Centralchem, Bratislava, Slovakia) and buffered with 0.165 M morpholinopropane sulfonic acid (MOPS, PanReac AppliChem, Darmstadt, Germany) to pH 7.0. Tested antifungals (fluconazole, FLU, Pfizer, New York, NY, USA; caspofungin, CAS, Merck Sharp & Dohme Ltd., London, UK; amphotericin B, AMB, Sigma Aldrich, Taufkirchen, Germany) were prepared in 2-fold serial dilutions ranging from 0.06–256 μg/mL for FLU, 0.015–16 μg/mL for CAS, and 0.06–8 μg/mL for AMB. Microtiter plates were incubated at 37 °C. After 24 h, optical density was measured at 570 nm using a plate reader (Dynex MRX-TC Revelation, Denkendorf, Germany). Susceptibility was evaluated in terms of minimal inhibitory concentration inhibiting the growth of the strain in the presence of the agent by 50% compared to the control sample without an antifungal agent (MIC_50_), and MIC_90_ representing the concentration that inhibits the growth of *C. auris* in the presence of AMB by 90% compared to the control sample without an agent. Each experiment was repeated at least 3 times with at least 4 parallel samples in each experiment.

### 2.2. Susceptibility Testing of FAR Alone and in Combination with FLU

Susceptibility testing was performed according to the EUCAST protocol as described above. The stock solution of 72 mM FAR (Sigma Aldrich, Taufkirchen, Germany) was prepared in 96% ethanol (Centralchem, Bratislava, Slovakia) and diluted to the following concentrations: 1 mM, 500 μM, 400 μM, 300 μM, 200 μM, 100 μM, 80 μM, 60 μM, and 50 μM in RPMI 1640 medium. The effectiveness of FAR was determined in terms of MIC_50_, as was previously described for antifungals. For determining the impact of FAR in combination with FLU, a subinhibitory concentration of FAR was tested (200 μM). Microtiter plates were seeded with 100 μL of the cell suspension (2 × 10^5^ cells/mL) and 50 μL of FAR was added to wells, and the plate was incubated at 37 °C for 1 h. Finally, 50 μL of 2-fold serial dilutions of FLU (ranging from 0.06 to 256 μg/mL) was added to the wells. The microtiter plates were incubated at 37 °C for 24 h and OD was measured (Dynex MRX-TC Revelation, Denkendorf, Germany). The results were evaluated in terms of change in MIC_50_ of FLU in combination with FAR. Each experiment was repeated at least 3 times with at least 4 parallel samples in each experiment. 

Drug–drug interactions were assessed by determining the fractional inhibitory concentration index (FICI) calculated according to the formula: FICI = FICI(A) + FICI(B), where FICI(A) = MIC(A) in combination/MIC(A) alone; FICI(B) = MIC(B) in combination/MIC(B) alone, where MIC(A) alone and MIC(B) alone are the MIC_50_ values of compounds A and B used alone and MIC(A) in combination and MIC(B) in combination are the MICs_50_ of compounds A and B at the effective combinations, respectively. The interaction between FLU and FAR was interpreted as synergistic when FICI was <0.5, as partially synergistic when FICI was ≥0.5 and <1, as an additive when FICI was 1, as indifferent when FICI was ≥ 1 and < 4, and as antagonistic when FICI was >4 [20,21].

### 2.3. qPCR Analyses of Genes Related to Resistance in the Presence of FLU and Combination FLU/FAR

At first, basal expression of genes related to the resistance (*CDR1, CDR2, MDR1,* and *ERG11*) was investigated by qPCR to determinate basal expression of tested genes. Briefly, total RNA was extracted from planktonic cells of *C. auris* harvested after overnight growth at YPD broth at 37 °C. Cells were washed twice with PBS and the pellet was prepared for isolation of RNA with GeneJET RNA Purification (ThermoScientific, Waltham, MA, USA) according to the manufacturer’s instructions. Then, we investigated the level of expression of the target genes after administration of FLU and FLU plus FAR. Total RNA was extracted as described above with slight modification in the yeast suspension preparation. *C. auris* isolates were grown overnight in YPD broth medium at 37 °C. After incubation, cells were centrifuged (5000× *g* for 5 min) and washed twice in sterile PBS and adjusted to concentration of 2 × 10^7^ cells/mL in 10 mL of sterile YPD broth (a) without compounds (control); and supplemented with (b) FLU—subinhibitory concentrations (*C. auris* H261—0.06 μg/mL; *C. auris* S—8 μg/mL; *C. auris* R—32 μg/mL); (c) combination FAR and FLU: FAR (200 μM) at 37 °C for 1 h and afterwards with FLU (same concentration mentioned above) at 37 °C for an additional 1 h. Post incubation, cells were pelleted (5000× *g* for 5 min) and given a sterile PBS wash. The washed cells were re-suspended in sterile PBS (1 mL) and the suspension was used for RNA extraction. RNA samples were treated with DNase I, RNase-free (Thermo Scientific, Waltham, MA, USA) to prevent contamination with genomic DNA. The cDNA was synthesized using a Maxima First Strand cDNA Synthesis kit for RT-qPCR (Thermo Scientific, Waltham, MA, USA) according to the manufacturer’s instructions and stored at −20 °C until use. The PCR primers used to amplify and identify the *C. auris* genes *CDR1, CDR2, MDR1, ERG11,* as well as *ACT1* genes are summarized in the Appendix A. All primers were synthesized by Metabion International AG (Planegg/Steinkirchen, Germany). For qPCR reaction, cDNA samples transcribed from 200 ng of total RNA were mixed with 5xHOT FIREPol^®^ EvaGreen^®^ qPCR Mix Plus (ROX) (Solis BioDyne, Tartu, Estonia) and amplified by using an Agilent Mx3000P qPCR System (Agilent Technologies, Inc., Santa Clara, CA, USA). Cycling conditions were as follows: one cycle of 15 min at 95 °C; followed by 40 cycles of 20 s at 95 °C and 30 s at 55 °C, followed by one cycle of 30 s at 72 °C for all genes. After amplification, a melting curve was run to ensure the absence of primer dimers. 

The level of gene expression was calculated using the 2^−ΔΔ^ CT method with respect to the housekeeping gene *ACT1.* Samples were compared to the control, which was represented by the most susceptible strain *C. auris* H261 cultivated without FLU or a combination FLU and FAR and normalized to 1. Each experiment was repeated at least three times with 3 parallel samples in each experiment.

### 2.4. Rhodamine 6G Intracellular Accumulation Assay and Fluorescence Microscopy

Intracellular accumulation assay was performed according to the protocol by Srivastava and Ahmad [5], with minor modification. Briefly, *C. auris* isolates were grown overnight in YPD broth medium at 37 °C. Then, *C. auris* suspensions were incubated with FLU and FAR as previously described in Section 2.3. Post incubation, cells were pelleted (5000× *g* for 5 min) and washed in sterile PBS. Cells were re-suspended in sterile PBS (1 mL) supplemented with 2% glucose and 4 μM rhodamine 6G (Sigma, Taufkirchen, Germany) and incubated at 37 °C for 30 min. Cells were then washed twice with cold sterile PBS and 1 mL of fresh PBS was added to the pellet. Afterwards, 100 μL of suspension was pipetted into a flat-bottomed dark 96 well plate (Costar^®^, Kennebunk, ME, USA) and fluorescence was measured with a fluorescence spectrophotometer (Tecan, Männedorf, Switzerland). The results were evaluated using MagellanTM Data Analysis Software and the intensity of fluorescence of samples was determined by relative fluorescence units (RFUs). The same suspension was immediately used for microscopy. Fluorescence was detected by an inverted fluorescence microscope (Zeiss, Jena, Germany) with excitation/emission spectra 525/548 nm. The pictures were captured by AxioCam ERc5s (Zeiss, Jena, Germany) and evaluated by software Motic Images Plus 3 (Hong Kong, China).

### 2.5. Statistical Analysis

Results were evaluated by statistical analysis using a one-way *t*-test in GraphPad Prism software (GraphPad, San Diego, CA, USA). Differences were considered statistically significant at *p* < 0.05 (*), *p* < 0.01 (**), *p* < 0.001 (***).

## 3. Results and Discussion

### 3.1. Identification of C. auris and Antifungal Susceptibility Profile of Tested Isolates

*C. auris*, also known as “the yeast superbug”, has become a serious emerging fungal pathogen since 2009 [1,22], but it might have emerged much earlier. Indeed, *Candida haemuloni* isolates from South Korea from 1996 have been re-identified and confirmed as *C. auris*, which suggested a high similarity between these two species [23]. Due to this observation, it is very important to use appropriate identification methods to correctly discriminate between *C. auris* and other *Candida* spp., mainly because of the increased resistance of this yeast to commonly used antifungals [22]. *C. auris* isolates from this study were controlled by phenotype and molecular analysis to avoid possible contamination (summarized in the Appendix A). The susceptibility profile was evaluated by determining the minimal inhibitory concentration (MIC) by a microdilution method to selected representatives of major classes of antifungals (FLU, CAS, and AMB). According to tentative breakpoints for *C. auris* published by the Center for Disease Control and Prevention (CDC; Atlanta, Georgia, USA) (https://www.cdc.gov/fungal/candida-auris/c-auris-antifungal.html, accessed on 29 May 2020), *C. auris* H261 manifested susceptibility to all tested antifungals (MIC_50_ rates: FLU = 0.125 μg/mL and CAS = 0.125 μg/mL; MIC_90_ rate: AMB = 0.5 μg/mL); *C. auris* S was susceptible to CAS (MIC_50_ = 0.125 μg/mL), resistant to AMB (MIC_90_ = 8 μg/mL), and showed decreased susceptibility to FLU (MIC_50_ for FLU = 16 μg/mL). *C. auris* R was resistant to all the three classes of antifungals (MIC_50_ rates: CAS = 8 μg/mL; FLU = 256 μg/mL, and MIC_90_ rate for AMB = 8 μg/mL). Our results agree with previously published data described for these *C. auris* isolates [17,18].

As was shown, the susceptibility profile of *C. auris* isolates used in this study was different and varied among isolates. Resistance to azoles has been reported in up to 91% of *C. auris* isolates [24]. In our study, *C. auris* S and *C. auris* R manifested a decreased susceptibility or resistance to FLU, whereas *C. auris* H261 remained susceptible to FLU as well as to both CAS and AMB. Susceptibility to azoles in *C. auris* is generally rare, but can be observed in healthy competent hosts that could be asymptomatically colonized with *C. auris* [17,25]. This is also the case of *C. auris* H261, which was isolated from the ear canal of an otherwise healthy man. On the other hand, azole resistance in *C. auris* remains a major problem in the treatment of candidiasis [9,24]. Some published works have already suggested that resistance could be associated with the regulation of genes involved in efflux or ergosterol biosynthesis [11,26]. Additionally, the resistance to multiple antifungal classes, as observed in *C. auris* R, is a phenomenon already reported [27,28,29,30,31,32].

### 3.2. Relative Expression Levels of Genes Related to the Azole Resistance (CDR1, CDR2, MDR1, and ERG11) in Untreated C. auris Isolates and after FLU and FAR Exposure

The overexpression of genes coding for the efflux pumps belonging to the ABC (*CDR1*, *CDR2*) and MFS (*MDR1*) superfamilies, together with the overexpression or point mutations of the *ERG11* gene, are the main mechanisms responsible for azole resistance in *Candida* spp. [33]. The genome of *C. auris* contains genes coding for at least 20 transmembrane domains associated with ABC transporters that could potentially contribute to azole resistance [11].

Expression of the above-mentioned genes (*CDR1*, *CDR2*, *MDR1*) and the *ERG11* gene was analyzed in this study (Figure 1). Results showed a significantly increased regulation of the *MDR1* gene in isolates of *C. auris* R and *C. auris* S (2.6 times and 19.6 in times increased, respectively), which was in agreement with previous observation indicating a resistance profile of both R and S isolates compared to the susceptible isolate H261. The efflux pump Mdr1p is a member of the MFS transporters responsible for the phenomenon of multiresistance in *C. albicans* [34,35]. Recently, Rybak et al. (2019) for the first time proved contribution of *MDR1* and *CDR1* to azole resistance in *C. auris* as well [26]. Our results agreed with this observation, confirming a native overregulation of the *MDR1* gene in resistant *C. auris* isolates, despite the observation that the level of up-regulation of the *MDR1* gene did not directly correspond to the level of the determined MIC_50_ to FLU in *C. auris* R. This indicated the participation of another mechanism of resistance against FLU as well.

Interestingly, the level of native expression of the *CDR* genes was not significantly changed in bothl resistant isolates; the 2.3 times higher abundance was observed only in the case of *C. auris* S for the *CDR1* gene. However, immediately after adding a subinhibitory concentration of FLU to both *C. auris* S and *C. auris* R, a significant up-regulation of the *CDR1* gene was observed (Figure 4A). These results suggested an activation of the efflux pumps and confirmed the overexpression of the gene for the efflux pump *Cdr1* as the major mechanism involved in *C. auris* resistance.

The expression of the *ERG11* gene was down-regulated in both *C. auris* S and *C. auris* R (1.7 times and 5.5 times less, respectively) compared to the susceptible *C. auris* H261, suggesting possible changes in lipid homeostasis [36,37]. This down-regulation was particularly pronounced in *C. auris* R which could significantly contribute to resistance of this isolate. In a recent study, Shahi et al. (2020) performed a detailed lipidomic analysis of *C. auris*. The authors suggested that composition of lipids differs among isolates of *C. auris* and seems to affect their resistance profile. They found a lower content of intermediate compounds, such as squalene, lanosterol, fecosterol, episterol, and fungisterol formed in the earlier stage of the ergosterol pathway in resistant isolates [25]. This finding could support our observation of a decreased expression of the *ERG11* gene in both *C. auris* R and S. Indeed, a down-regulation of the *ERG 11* gene reflects a reduction in target protein production and can result in a decrease in the amount of lanosterol, as observed in the work of Shahi et al. (2020), that in turn can affect FLU efficacy [25]. Additionally, the down-regulation of the *ERG11* gene in resistant isolates could also be triggered by mutations in this gene, another mechanism mediating resistance to azoles in *C. auris* [38,39].

### 3.3. The Synergy between FAR and FLU Modulates C. auris Resistance to FLU 

The combination of FAR with common antifungals has already been described by many authors in *C. albicans* [14,40,41,42], but only a few studies discussed the potential use of FAR in *C. auris* [5,6,16]. The main part of this work was focused on the impact of the quorum-sensing molecule FAR on FLU-resistant *C. auris* isolates (*C. auris* S and *C. auris* R) compared to the susceptible isolate *C. auris* H261. Our results confirmed a partially synergistic effect between FAR and FLU, with a FIC index of 0.75 and 0.4 for *C. auris* S and *C. auris* R, respectively, that was supported by changes in expression of already mentioned resistance genes shown in the next paragraph. These results pointed to the possible role of FAR as a modulator of efflux pumps in resistant *C. auris* isolates.

At first, MIC_50_ of FAR alone was tested by a microdilution method. The susceptibility of *C. auris* isolates was tested for 11 different concentrations of FAR. Results showed different rates of MIC_50_ for FAR among isolates (*C. auris* H261 = 500 μM, *C. auris* S = 400 μM, and *C. auris* R = 500 μM). Results are illustrated in Figure 2. Interestingly, MIC_50_ of FAR alone was generally lower in all the tested *C. auris* isolates compared *C. albicans*, which displayed a MIC_50_ for FAR of 1 mM [14]. This observation agreed with Nagy et al. (2020), who described the MIC_50_ of FAR to be slightly lower, i.e., 300 μM, for *C. auris* [6]. In addition, FAR susceptibility variations might be strain-dependent and could be influenced by the amount of intracellular FAR as well. As FAR is a substrate of the Cdr1 pump in *C. albicans* [43], we hypothesize that differences in the FAR MIC_50_ among tested *C. auris* isolates could rely on the level of the native expression of the *CDR1* gene. 

In the previous paragraph, the difference in expression of the genes related to resistance in *C. auris* isolates was described. Concretely, isolate *C. auris* S manifested the highest abundance of the *CDR1* gene (2.35 times higher compared to *C. auris* H261) that could lead to a higher inhibitory effect of FAR alone on this isolate compared to *C. auris* H261 and *C. auris* R. With higher activity of the *CDR1* gene, FAR could accumulate more in the cells, resulting in lower MIC for FAR in *C. auris* S, which was also found to be less susceptible to FLU. These results could be supported by Zamith-Miranda et al. (2018) who found higher expression of the gene coding for efflux transporter Cdr1 in FLU-resistant *C. auris* compared to FLU-susceptible *C. albicans* [37].

A subinhibitory concentration of FAR (200 μM) was selected to be tested in combination with FLU (Figure 3, gray columns). In the case of FLU-susceptible strain *C. auris* H261, FAR did not reduce FLU MIC_50_ values and the FICI was indifferent (Figure 3A). Changes in MIC_50_ were observed for both *C. auris* S (Figure 3B) and *C. auris* R (Figure 3C). A partial synergy was observed in *C. auris* S (the MIC_50_ for FLU decreased from 16 to 4 μg/mL), and a synergy was observed in *C. auris* R (the MIC_50_ for FLU decreased from 256 to 1 μg/mL). Results are summarized in Table 1. Our data are in agreement with the report of Nagy et al. (2020), confirming a synergism between FAR and voriconazole, posaconazole, itraconazole, and FLU, although the concentration of FAR used in the combination was lower (75 μM) compared to our data [6]. Interestingly, the same authors described a potential synergy of FAR in combination with echinocandins in *C. auris* [6].

### 3.4. FAR Down-Regulates the Expression of Genes Involved in Resistance to FLU

Recently published studies described the potential of FAR as an inhibitor of efflux pumps in resistant *C. albicans* [10,14]. Until now, only one report hypothesized a similar effect in *C. auris* [5], but the expression of genes was not investigated. qPCR was performed to investigate possible changes in the *CDR1, CDR2, MDR1,* and *ERG11* expression after FAR plus FLU treatment in all the *C. auris* isolates. 

We found an up-regulation of the *CDR1* gene in both *C. auris* S and *C. auris* R (Figure 4A,B) treated with FLU. The relative fold change for *CDR1* and *CDR2* genes was higher in *C. auris* S (2.9 and 1.28 times for the *CDR1* and *CDR2* gene, respectively) and *C. auris* R (2.09 and 1.97 times for the *CDR1* and *CDR2* gene, respectively) compared to *C. auris* H261 (control). Our results agreed with the work of Wasi et al. (2019), proving that azole-resistant *C. auris* displays the overexpression of the *CDR1* gene resulting in the efflux of FLU outside the cell [11]. On the contrary, we did not detect a significant change in the *MDR1* expression after the addition of FLU to both resistant isolates of *C. auris* (Figure 4C). Accordingly, Hiller et al. (2006) showed a less efficient transport of FLU by Mdr1p in *C. albicans* [44]. In conclusion, our data support previous studies suggesting that a response of *C. auris* to the presence of FLU is mainly mediated by an efflux pump encoded by the *CDR1* gene [7,9,11].

Afterwards, we studied changes in the gene expression after a combined treatment with FAR and FLU. A decrease in the expression of both the *CDR1* and *CDR2* genes was observed in resistant *C. auris* isolates (Figure 4A,B), suggesting an inhibitory effect of FAR on efflux transporters, but results proved to be significant only for the *CDR1* gene.

The modulatory effect of FAR on the efflux pumps *Cdr1* and *Cdr2* was described for the first time in *C. auris* by Srivastava et al. in 2020 [5]. Our results agreed with that finding despite the concentration of FAR used in our experiments being much lower (200 μM) compared to that used by the mentioned authors (125 mM). We hypothesized that effectiveness of FAR on *C. auris* depends on up-regulation of the *CDR1* gene. This claim was supported by observation that pointed to FAR’s ability to induce apoptosis in *C. albicans* that was mediated by the Cdr1 pump resulting in extrusion and depletion of intracellular glutathione [43,45]. In agreement with this conception, the basal expression of the *CDR1* gene in *C. auris* S was higher compared to susceptible isolate *C. auris* H261. However, this was not confirmed in the most resistant isolate *C. auris* R. On the other hand, regardless of the basal data, FLU increased the *CDR1* gene expression in both isolates. It is suggested that FLU might also activate the *CDR1* gene in combination with FAR. Subsequently, FAR could effectively bind the Cdr1 pump to enter the cell, inhibit efflux, and help FLU be restored inside the cell. The presented results confirmed a synergy of FAR and FLU resulting in the increase in susceptibility of *C. auris* S and *C. auris* R resistant to FLU.

Additionally, a significant decrease in the *ERG11* gene expression after administration of FAR in combination with FLU (Figure 4D) was observed in both *C. auris* S and *C. auris* R. The modulatory effect of FAR in the regulation of ergosterol synthesis has been described in *C. albicans* by Yu et al. (2012). Similarly, the down-regulation of the *ERG1, ERG3, ERG6, ERG11,* and *ERG25* genes was observed in *C. albicans* biofilm treated with FAR [46]. The *ERG25* gene (coding for methylsterol monooxygenase) and the *ERG4* gene (coding for delta 24(24(1))-sterol reductase) were both shown to be down-regulated similarly to the FAR-exposed group of *C. albicans* described in the study of Wang and Liu (2019). It was also concluded that exogenous FAR has an evident, but non-deterministic, effect on the synthesis of ergosterol in *C. albicans* [47]. These results might suggest an indirect effect of FAR on ergosterol biosynthesis, although the mechanism underlying this ability is still largely unclear. Overexpression of the *ERG11* gene is also known for playing a role in *C. auris* azole resistance [38,48]. The FAR-dependent down-regulation of the *ERG11* gene in resistant strains of *C. auris* increased the inhibitory effect of FLU and could be a promising tool to overcome azole resistance. 

Finally, changes in the *MDR1* gene expression have not been observed in combination of FAR and FLU in resistant *C. auris* isolates (Figure 4C). From the point of view of FAR’s effect, the contribution of Mdr1 is not relevant in this study, due to the fact that FAR is not the substrate of this efflux transporter [43,45]. 

### 3.5. FAR Inhibits the Cdr1 Pump Allowing for a High Intracellular Accumulation of Rhodamine 6G 

Results from the previous experiment suggested high activity of FAR as a potential modulator of the *CDR1* and *ERG11* gene expression related to azole resistance and the ergosterol pathway as well. To confirm the inhibitory effect of FAR on the activity of the *C. auris* Cdr1 efflux pump, the intracellular accumulation of rhodamine 6G dye (R6G) was studied in treated and untreated isolates. Untreated *C. auris* isolates were used as a control (Figure 5A, dark gray columns). FLU treatment alone resulted in a signal similar to that observed in the control group, confirming an active efflux (Figure 5A, light gray columns). The highest fluorescence signal has been observed in *C. auris* isolates treated with the FAR and FLU combination (Figure 5A, black columns). These results suggest a FAR-dependent inhibition of Cdr1 that blocks the efflux and restores the intracellular FLU. This hypothesis is supported by fluorescent microscopy. Indeed, there was a marked increase in the R6G fluorescence signal in cells treated with the combination of FAR and FLU compared to the untreated control or sample treated with FLU alone (Figure 5B, *C. auris* R). The same results were observed in both isolates *C. auris* H261 and *C. auris* S. These results corroborate qPCR data concerning the inhibitory effect of FAR on the *CDR1* gene in combination with FLU. R6G is substrate for the Cdr1 pump in *C. albicans,* and it is very effective in the measurement of activity of this efflux transporter pump [49]. Srivastava et al. (2020) confirmed for the first time inhibition of Cdr1 with FAR in *C. auris* [5]. Our results are in agreement with that observation, but our study confirmed the effectivity of FAR and FLU synergy by molecular analysis. FAR in combination with FLU helps to accumulate FLU in the cells and blocks efflux through regulation of the *CDR1* gene resulting in an increase in susceptibility of resistant *C. auris* isolates. 

## 4. Conclusions

Due to the strong impact of efflux in *C. auris* azole resistance, inhibitors/modulators of efflux pumps seem to represent a promising tool to overcome multiresistance of this yeast. FAR is a quorum-sensing molecule with a high potential to modulate efflux by inhibiting the Cdr1 transporter. Besides the inhibitory effect of FAR alone, the subinhibitory concentration of FAR in combination with FLU significantly decreased the expression of the *CDR1* gene in agreement with a previously manifested shift of MIC_50_ for FLU. However, observed changes were shown to be *C. auris* strain-dependent, probably associated with a native expression of the studied genes. Additionally, the presented study proved for the first time that FAR contributed to decreased regulation of the *ERG11* gene resulting in possible disturbance of lipid homeostasis. In summary, the presented data confirmed a promising role of FAR that can modulate resistance to azoles in resistant *C. auris* isolates and showed promising activity in combination with FLU.

## Figures and Tables

**Figure 1 jof-08-00783-f001:**
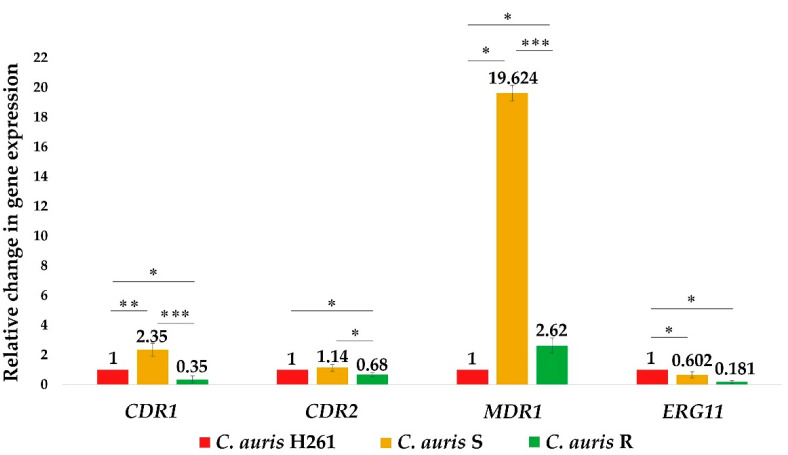
Basal expression of the *CDR1, CDR2, MDR1,* and *ERG11* genes in *C. auris* isolates. The relative expression was calculated by qPCR using the 2^−ΔΔ^ CT method with respect to the housekeeping gene *ACT1* and using the susceptible isolate *C. auris* H261 for normalization. Data represent the average of 3 independent experiments performed in triplicate ± SD. *p*-values < 0.05 were considered significant: * <0.05; ** < 0.01; *** <0.001.

**Figure 2 jof-08-00783-f002:**
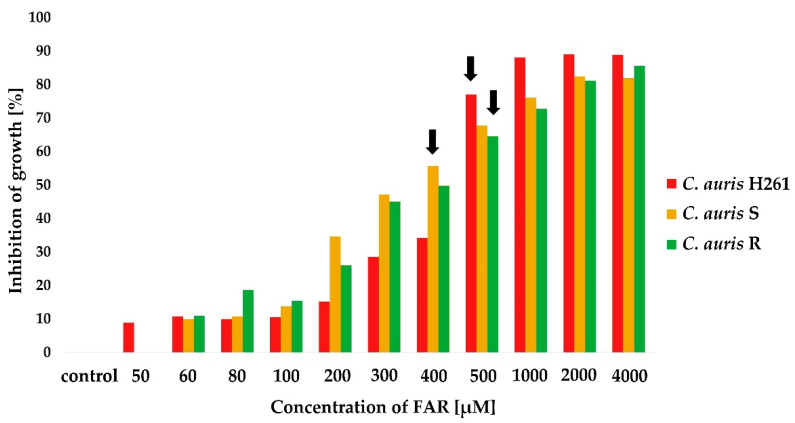
Inhibitory effect of FAR on the planktonic cells of *C. auris* determined by MIC_50_ (labeled with black arrows). Optical density (OD_580_) of suspension was determined after 24 h cultivation of yeast in the presence of different concentrations of FAR; the control sample was without FAR. Percentage of growth inhibition was calculated from the OD_580_ values of samples compared to the inhibition of the control sample set to 0%. Data represent the average of 3 independent experiments performed in triplicate.

**Figure 3 jof-08-00783-f003:**
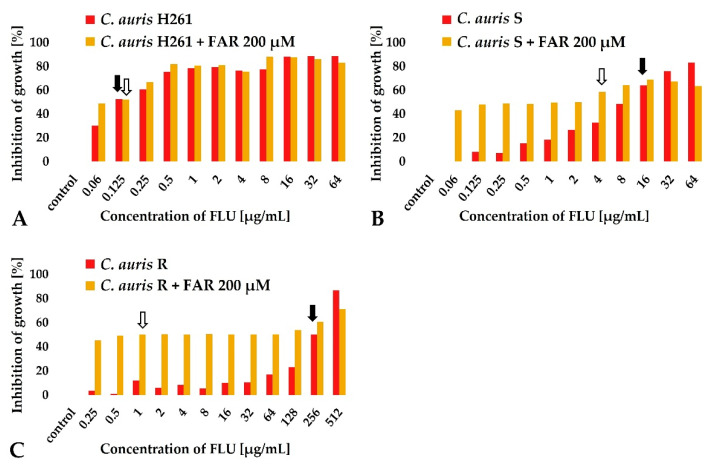
Synergistic effect of FAR (200 μM) in combination with FLU determined by changes in MIC_50_ of FLU. MIC_50_ values of FLU alone are marked with a black arrow. MIC_50_ of FLU combined with FAR are marked with a white arrow. MIC_50_ of FLU remained without change for *C. auris* H261 (**A**), MIC_50_ of FLU shifted from 16 μg/mL to 4 μg/mL for *C. auris* S (**B**), MIC_50_ of FLU shifted from 256 μg/mL to 1 μg/mL for *C. auris* R (**C**). Data represent the average of 3 independent experiments performed in triplicate.

**Figure 4 jof-08-00783-f004:**
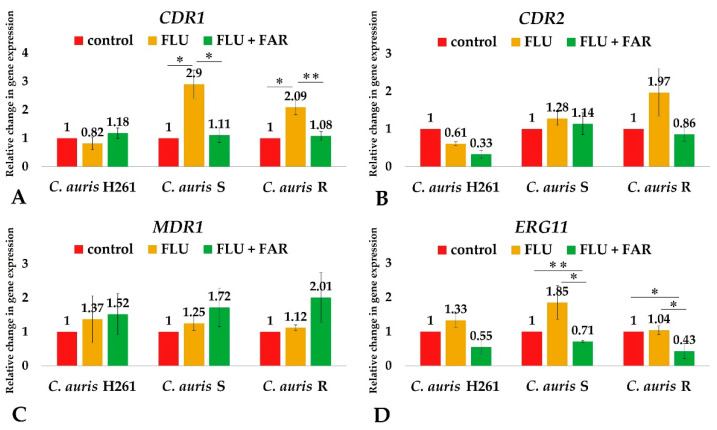
Relative change expression of the *CDR1* (**A**), *CDR2* (**B**), *MDR1* (**C**), and *ERG11* (**D**) genes determined by qPCR. The expression of genes was calculated using the 2^−ΔΔ^ CT method with respect to the housekeeping gene *ACT1*. The level of gene expression was examined in the presence of subinhibitory concentration of FLU: 0.06 μg/mL for *C. auris* H261, 8 μg/mL for *C. auris* S, and 32 μg/mL for *C. auris* R (the second column in each chart); a combination of 200 μM FAR and subinhibitory concentration of FLU (the third column in each chart). Values were compared to the control without any drugs (the first column in each chart) which was normalized to a value of 1. Data represent the average of 3 independent experiments performed in triplicate ± SD. *p*-values < 0.05 were considered significant: * < 0.05; ** < 0.01.

**Figure 5 jof-08-00783-f005:**
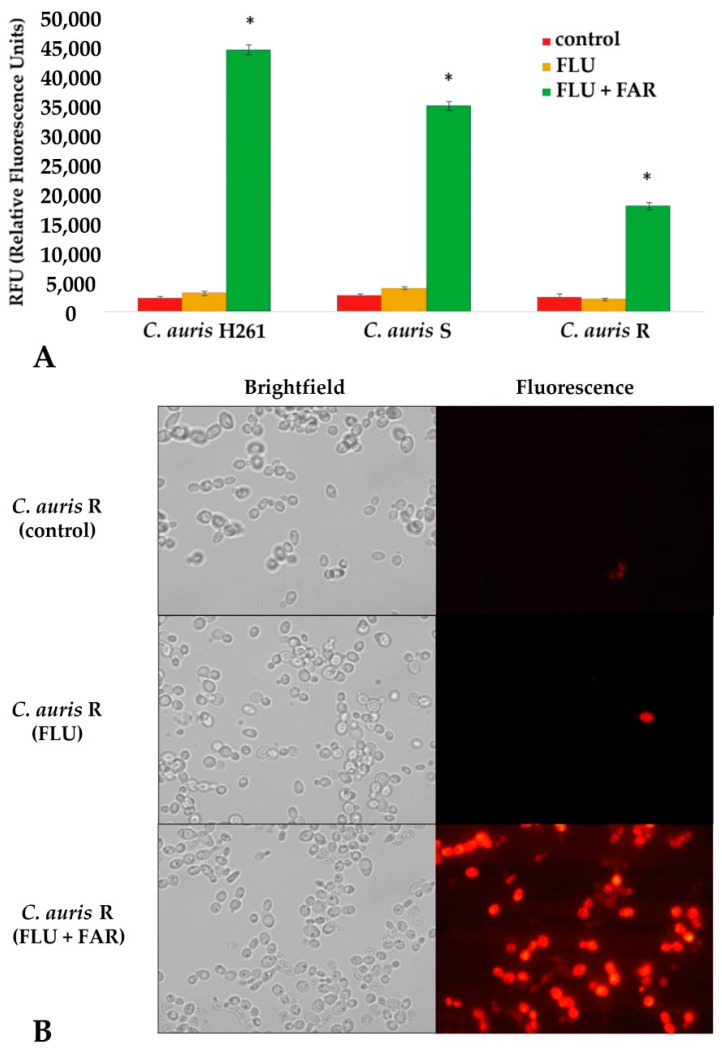
(**A**) Rhodamine 6G intracellular accumulation measured by relative fluorescence. The figure shows a significant increase in the accumulation of R6G dye in all *C. auris* treated with a combination of FAR and FLU resulting in the highest abundance of fluorescence (green columns). The cells treated only with the subinhibitory concentration of FLU (yellow columns) have the same level of R6G as control cells without any agent (red columns) which means that R6G was extruded from cells by efflux. Subinhibitory concentrations of FLU were as follows: 0.06 μg/mL for *C. auris* H261, 8 μg/mL for *C. auris* S, and 32 μg/mL for *C. auris* R. (**B**) Fluorescence microscopy of the selected yeast suspension of the most resistant isolate *C. auris* R confirms the highest accumulation of R6G in the cells after a combination of FAR and FLU, confirming FAR as an inhibitor of efflux pump Cdr1; *p*-values * < 0.05 were considered significant.

**Table 1 jof-08-00783-t001:** Synergy between FLU and FAR (200 μM) of *C. auris* isolates determined according to MIC_50_ values.

Strain	MIC_50_ FLU (Alone) (μg/mL)	Interpretation	MIC_50_ FAR (Alone) (μM)	MIC_50_ FLU with 200 μM FAR (μg/mL)	FIC Index	Interpretation
*C. auris* H261	0.125	Susceptible	500	0.125	1.4	Indifferent
*C. auris* S	16	Susceptible *	400	4	0.75	Partial synergy
*C. auris* R	256	Resistant	500	1	0.4	Synergy

FLU—Fluconazole; FAR—Farnesol; MIC_50_—Minimal inhibitory concentration of tested antifungals inhibiting the growth of yeast cells by 50% compared to the control sample without an agent. * Isolate *C. auris* S is susceptible according to the tentative breakpoint value for *C. auris*, which is 32 μg/mL (https://www.cdc.gov/fungal/candida-auris/c-auris-antifungal.html, accessed on 29 May 2020). However, MIC_50_ of 16 μg/mL is considered to be relatively high and means significantly decreased susceptibility of this isolate to FLU.

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
