# Peer review of "Farnesol Boosts the Antifungal Effect of Fluconazole and Modulates Resistance in Candida auris through Regulation of the CDR1 and ERG11 Genes"

_jof, 2022, doi:10.3390/jof8080783_

Round 1

Reviewer 1 Report

In the present study, the effect of farnesol (FAR) on the antifungal activity of fluconazole (FLU) against the human pathogen Candida auris was investigated. Previous studies with related fungal pathogens revealed an inhibition of drug efflux pumps by FAR, resulting in increased susceptibility against antifungal agents. In the current manuscript, a similar effect is observed for C. auris and gene expression analysis results support the idea that changes in the expression levels of genes encoding efflux pumps and the ergosterol biosynthesis gene ERG11 are functionally implicated in the changes of FLU susceptibility. 

The current study uses three different C. auris strains: H261 (FLU sensitive), S (reduced sensitivity) and R (resistant). FAR increased FLU susceptibility in the isolates S and R, but not in H261. As a possible explanation, CDR1, CDR2, MDR1 and ERG11 mRNA levels were measured in the different strains in presence and absence of FLU/FAR. The resistant isolates S and R showed FLU induced upregulation of CDR1, while changes for CDR2 do not appear statistically significant. Figure 4 lacks an explanation which p-values are indicated by * and **. Anyhow, changes for CDR2 are not marked by * or ** and the statement on line 339 indicates that CDR1 (but not CDR2) is upregulated after FLU exposure and this effect is prevented in presence of FAR. In the abstract and final conclusion of this results section (line 350f), however, the effect is ascribed to both, CDR1 and CDR2 which is not supported by the data shown.

Since strains S and R differ strongly in their MIC50, with strain R being much more resistant to FLU than strain S, the explanation that these phenotypes are mainly linked to changes in CDR1 (and CDR2) expression levels does not seem fully plausible because the relative change in CDR1 expression is greater in strain S as compared to strain R. The opposite would be expected if this is the main reason for FLU resistance in these isolates. This should be critically discussed and any extension to CDR2 carefully checked and removed if no statistically significant changes are observed.

Concerning the changes in untreated C. auris strains, the authors state (line 238): “Our results are in agreement with this observation, confirming a native high over-regulation of MDR1 in resistant C. auris isolates.” However, again, the relative change is not in line with the degree of FLU susceptibility: MDR1 is upregulated much more strongly in strain S as compared to strain R. If this would be the main reason for FLU phenotypes, the opposite would be expected. Hence, strain S, being more sensitive to FLU shows higher expression levels of MDR1 (in absence of FLU) and stronger FLU induced expression of CDR1 as compared to strain R, which is inconsistent with the much stronger FLU resistance phenotype of strain R.  These observations should be discussed more critically.

Since MDR1,CDR1 and ERG11 were previously shown to be involved in FLU resistance and the three strains used in this study show changes in mRNA levels for these genes that are not fully consistent with relative changes in FLU susceptibility, I would recommend to sequence all three loci (and possibly also the CDR2 gene) from the different isolates. It appears at least possible that mutations in these genes might contribute to FLU phenotypes observed. Also, if no mutations are present, this would represent a valuable additional information.

Author Response

We thank the reviewer for his time and evaluation of our manuscript.

Please, find enclosed our response.

Reviewer comments:

Reviewer #1: In the present study, the effect of farnesol (FAR) on the antifungal activity of fluconazole (FLU) against the human pathogen Candida auris was investigated. Previous studies with related fungal pathogens revealed an inhibition of drug efflux pumps by FAR, resulting in increased susceptibility against antifungal agents. In the current manuscript, a similar effect is observed for C. auris and gene expression analysis results support the idea that changes in the expression levels of genes encoding efflux pumps and the ergosterol biosynthesis gene ERG11 are functionally implicated in the changes of FLU susceptibility.

The current study uses three different C. auris strains: H261 (FLU sensitive), S (reduced sensitivity) and R (resistant). FAR increased FLU susceptibility in the isolates S and R, but not in H261. As a possible explanation, CDR1, CDR2, MDR1 and ERG11 mRNA levels were measured in the different strains in presence and absence of FLU/FAR. The resistant isolates S and R showed FLU induced upregulation of CDR1, while changes for CDR2 do not appear statistically significant. Figure 4 lacks an explanation which p-values are indicated by * and **.

Explanation about a level of significance (* or **) has been added in the legend of Figure 4.

Anyhow, changes for CDR2 are not marked by * or ** and the statement on line 339 indicates that CDR1 (but not CDR2) is upregulated after FLU exposure and this effect is prevented in presence of FAR. In the abstract and final conclusion of this results section (line 350f), however, the effect is ascribed to both, CDR1 and CDR2 which is not supported by the data shown.

Thank you for the notice. Yes, if the change is not significant, it can not be considered relevant. Therefore, we have changed the information in this meaning in the entire manuscript including Abstract and Conclusions.

Since strains S and R differ strongly in their MIC50, with strain R being much more resistant to FLU than strain S, the explanation that these phenotypes are mainly linked to changes in CDR1 (and CDR2) expression levels does not seem fully plausible because the relative change in CDR1 expression is greater in strain S as compared to strain R. The opposite would be expected if this is the main reason for FLU resistance in these isolates. This should be critically discussed and any extension to CDR2 carefully checked and removed if no statistically significant changes are observed.

As it was mentioned, the meaning of the Cdr2 pump has been modified in terms of the results achieved. The focus was on the CDR1 gene. However, since the CDR1 gene was upregulated in C. auris R only in the presence of FLU, this information was highlighted. In addition, the native regulation of the ERG11 gene was significantly reduced, while a very high down-regulation was observed only in the C. auris R isolate indicating a possible participation in resistance to FLU in this isolate. This was also added as a possible explanation of resistance in C. auris R. Supporting data were experimentally checked again and included in Figure 1.

Concerning the changes in untreated C. auris strains, the authors state (line 238): “Our results are in agreement with this observation, confirming a native high over-regulation of MDR1 in resistant C. auris isolates.” However, again, the relative change is not in line with the degree of FLU susceptibility: MDR1 is upregulated much more strongly in strain S as compared to strain R. If this would be the main reason for FLU phenotypes, the opposite would be expected. Hence, strain S, being more sensitive to FLU shows higher expression levels of MDR1 (in absence of FLU) and stronger FLU induced expression of CDR1 as compared to strain R, which is inconsistent with the much stronger FLU resistance phenotype of strain R.  These observations should be discussed more critically.

The discussion of results has been changed in suggested meaning of the reviewer.

Since MDR1,CDR1 and ERG11 were previously shown to be involved in FLU resistance and the three strains used in this study show changes in mRNA levels for these genes that are not fully consistent with relative changes in FLU susceptibility, I would recommend to sequence all three loci (and possibly also the CDR2 gene) from the different isolates. It appears at least possible that mutations in these genes might contribute to FLU phenotypes observed. Also, if no mutations are present, this would represent a valuable additional information.

We are sorry, but it is problematic to comply with the reviewer's request. The sequencing of more than 4 genes would require a lot of further experimental work, which we estimate to take half a year, including a complete comparative analysis, and would be a sufficient basis for another manuscript. Given that the main goal of the work was to describe the role of FAR in restoring resistance of C. auris, the sequencing of the tested genes would be interesting, but it would not affect the results in view of the effect of FAR on restoring of susceptibility of C. auris to FLU as the main topic of the manuscript. Thank you for the suggestion, we will consider it in our future work.

Reviewer 2 Report

I have no comments that could improve this paper

Author Response

Reviewer #2: I have no comments that could improve this paper

We thank reviewer for his opinion and revision of our manuscript.

Reviewer 3 Report

Dekkerová et al examined the effect of farnesol to fluconazole resistance and the expression of fluconazole resistance related genes. The reported findings are more or less well-known in case of C. albicans (as published previosly for example by this group too). However, the number of data dealing with C. auris related changes is limited. Generally, the manuscript is well-written, comprehensive and moderate. The results are clear and well explained. I have only some minor suggestions, which further improve the quality of manuscript:

Line 96 RPMI-1640 contained bicarbonate or not?

Line 109 How much % ethanol is found in intreated control cells?

Line 184 „Highly” and „Extremely” should be removed. The significant is significant.

Line 191 „Candida auris” should be abbreviated as C. auris

Line 273 „synergistic effect between FAR and FLU, with a FIC index of 0.75 and 0.4” Between 0.5 and 0.75 we don’t speak about synergistic just partially synergistic interactions. Please reword this sentence.

All Figures (bar charts) should colorize.

Author Response

We thank the reviewer for his time and evaluation of our manuscript.

Please, find enclosed our response.

Reviewer #3: Dekkerová et al examined the effect of farnesol to fluconazole resistance and the expression of fluconazole resistance related genes. The reported findings are more or less well-known in case of C. albicans (as published previosly for example by this group too). However, the number of data dealing with C. auris related changes is limited. Generally, the manuscript is well-written, comprehensive and moderate. The results are clear and well explained. I have only some minor suggestions, which further improve the quality of manuscript:

  1. Line 96 RPMI-1640 contained bicarbonate or not?

We used the RPMI-1640 without bicarbonate.

  1. Line 109 How much % ethanol is found in intreated control cells?

We added concentration of stock solution of FAR (72 mM) in ethanol as well as information that serial dilution of FAR was done in RPMI medium. Therefore, amount of ethanol was negligible in tested samples.

  1. Line 184 „Highly” and „Extremely” should be removed. The significant is significant.

We accepted note of reviewer and removed a discrimination in a definition of significance, but generally, level of significance is given by statistic calculation. In any way, results with p0.001 (***) have higher value then those on level p0.5 (*).

  1. Line 191 „Candida auris” should be abbreviated as C. auris

It was changed

  1. Line 273 „synergistic effect between FAR and FLU, with a FIC index of 0.75 and 0.4” Between 0.5 and 0.75 we don’t speak about synergistic just partially synergistic interactions. Please reword this sentence.

It was changed according to recommendation.

  1. All Figures (bar charts) should colorize.

Figures was changed and uploaded in color version.

Round 2

Reviewer 1 Report

The authors adjusted conclusions concerning the FAR effect on the expression of CDR2 (which was non-significantly dowregulated). I strongly suggest to also change the title of the paper (remove CDR2 there).

Sequencing of the CDR1/2 and ERG11 genes would have been informative but the argumentation of the authors to explain why this could not be done in due time seem reasonable.